# Load–Temperature Coupling Effect on the Base Plate End of the Whole Tram Road

**Chongwei Huang, Meixuan Zhu and Yu Sun** *

Department of Transportation Engineering, University of Shanghai for Science and Technology, Shanghai 200093, China; hcwei@126.com (C.H.); zmxuan57@126.com (M.Z.)
* Correspondence: ysun@usst.edu.cn

**Abstract:** Although trams have been widely recognized, systematic and comprehensive research on their design and construction is lacking. Based on the ABAQUS finite element software, we constructed a three-dimensional finite element analysis model of the overall track bed of the tram. Taking the most unfavorable working condition of load and temperature coupling as the research object, that is from 5:30 to 6:00 a.m., the load was applied to the plate end position. The simulation experiments were carried out by selecting different thicknesses of the track bed slab, support layer thickness, contact conditions between the track bed slab and the support layer, the modulus of the track bed slab, the modulus of the support layer and the soil foundation strength, and the stress and deflection of the subgrade were calculated. The most unfavorable load–temperature coupling condition was taken as the research object, that is, applying a load of 5.5–6 points on the plate end. Different track bed slab thicknesses, support layer thicknesses, contact conditions between track bed slab and support layer, track bed slab moduli, support layer moduli, and foundation strengths were utilized to conduct simulation tests for calculating the stress and deflection of the subgrade. Under the coupling effect of load on the end of the slab and the effect of temperature, changing the thickness of the track bed slab and the coefficient of friction between layers can improve the lateral force and deflection of the track bed slab. The effect of deflection is small. Changing the thickness of the support layer has an insignificant effect on the stress on the top surface of the soil foundation and the deflection of the top surface of the subgrade. The modulus of the track bed slab can affect the lateral force and deflection of the track bed slab, but it only slightly affects the longitudinal force and deflection of the track bed slab and the longitudinal and lateral force and deflection of the soil foundation. The modulus of the supporting layer only slightly affects the vertical and horizontal force and deflection of the track bed slab and soil foundation. The soil foundation modulus has the greatest influence on the vertical and horizontal forces and deflection of the track bed slab and soil foundation.

**Keywords:** tram; roadbed; finite element method; load and temperature; coupling effect

## 1. Introduction

A tram is a light rail transit vehicle that uses electricity to drive and travel on a track [1–3]. Tramcars have experienced four stages of development: rise, peak, decline, and revival. They have the advantages of high safety factor, low noise, relatively low engineering cost, and high environmental performance [4–6]. Research on modern trams mainly focuses on issues such as traffic organization, connection with other modes of transportation, replacement of power supply systems, landscape impact, and acoustics [7,8]. The stress state of tram subgrade under load has rarely been examined [9]. Shamalta and Metrikine explored the mechanical response of the embedded railway track based on Fourier integral transform and they compared the research results of a two-dimensional model and a simplified one-dimensional model [10]. Ling established a three-dimensional simulation model of the tram embedded track system based on multi-body dynamics and

finite element analysis methods [11]. Using the track subgrade three-dimensional finite element model, Guo provided the mapping relationship between the differential settlement of the subgrade and the unevenness of the tram track. Then, he used the tram vehicle–track coupling model to estimate the effect of subgrade settlement on the vehicle–rail interaction [12]. However, the abovementioned scholars ignored the different environmental temperature conditions, which greatly influences the stress on the roadbed [13–15]. Zhang Chuanfeng, Song Hongfang, and Wang Ruiji mainly calculated the frost heave and mechanical behavior of the whole roadbed under the effect of temperature [16–18]. Japanese scholars used a two-dimensional finite element model to analyze the daily temperature difference, seasonal temperature difference, and thermal stress [19]. Some scholars have tested the influence of the temperature stress of the bottom plate on the structural design [20–22], and they have established a calculation method for the temperature stress of the bottom plate [23,24]. Finite element analysis is a better method for pavement design in the research and experiment of the temperature stress calculation of cement concrete pavement [25–27]. In summary, most scholars have established the relationship between temperature and load on the basis of the Winkler foundation. Although some scholars can simulate the coupling effect of temperature field and stress field, the tram model structure is relatively simple, and the mechanical behavior analysis of the overall subgrade considering the thickness of the structural layer and material parameters is lacking.

Therefore, in this study, a three-dimensional finite element model is established based on ABAQUS, and the most unfavorable working condition under load–temperature coupling is taken as the research object. Through finite element calculation, the influence of different track bed slab thickness, support layer thickness, contact conditions between track bed slab and support layer, track bed slab modulus, support layer modulus, and soil foundation strength comprehensively consider the mechanical properties of a single pavement base. The research results have important guiding significance for tram engineering design.

## 2. Theory

### 2.1. Calculation Method

The temperature fatigue stress generated at the critical load position of the double-layer slab surface of the elastic foundation and the maximum temperature stress of the concrete surface slab at the maximum temperature gradient are as follows:

$$
\begin{cases}
\sigma_{tr} = k_t \sigma_{t,\max} \\
\sigma_{t,\max} = \dfrac{\alpha_c E_c h_c T_g}{2} B_L \\
k_t = \dfrac{f_r}{\sigma_{t,\max}} \left[ a_t \left( \dfrac{\sigma_{t,\max}}{f_r} \right)^{b_t} - c_t \right]
\end{cases}
\qquad
\begin{cases}
B_L = 1.77 e^{-4.48 h_c} C_L - 0.131(1 - C_L) \\
C_L = 1 - \left( \dfrac{1}{1+\xi} \right) \dfrac{\sinh t \cos t + \cosh t \sin t}{\cos t \sin t + \sinh t \cosh t} \\
t = \dfrac{L}{3 r_g} \\
\xi = - \dfrac{\left( k_n r_g^4 - D_c \right) r_\beta^3}{\left( k_n r_\beta^4 - D_c \right) r_g^3} \\
r_\beta = \left( \dfrac{D_c D_b}{(D_c + D_b) k_n} \right)^{0.25} \\
k_n = \dfrac{1}{2} \left( \dfrac{h_c}{E_c} + \dfrac{h_b}{E_b} \right)^{-1}
\end{cases}
\tag{1}
$$

where $\sigma_{tr}$ is the temperature fatigue stress at the critical load of the facing plate; $\sigma_{t,\max}$ is the maximum temperature stress generated by the surface laminate during the maximum temperature gradient; $k_t$ is the temperature fatigue stress coefficient considering the cumulative fatigue effect of temperature stress; $\alpha_c$ is the linear expansion coefficient of concrete; $T_g$ is the maximum temperature gradient in 50 years where the road is located; $B_L$ is the temperature stress coefficient of the combined temperature warpage stress and internal stress; $a_t$, $b_t$, and $c_t$ are regression coefficients; $\xi$ is a parameter related to the double-layer board structure; $r_\beta$ is the parameter of the contact condition between layers; $k_n$ is the vertical contact stiffness between the surface and base layers. When an asphalt concrete interlayer or isolation layer is set between the two layers, it is taken as 3000 MPa/m; $h_c$, $D_c$, and $E_c$ are the thickness of the upper plate, the flexural stiffness of the section and the flexural modulus of elasticity, respectively; $h_b$, $D_b$, and $E_b$ are the thickness of the lower

plate, the flexural stiffness of the section and the flexural modulus of elasticity, respectively. $L$ is the lateral spacing of the cladding panels.

### 2.2. Boundary Conditions

The boundary conditions of the monolithic roadbed of the tram are restricted by two methods. One is the relationship between the heat flow through the cement surface, surface temperature ($T$), air temperature ($T_a$), and solar radiation as follows:

$$-\lambda \frac{\partial T}{\partial n} = \beta(T - T_a) - \alpha_S S \tag{2}$$

where $\lambda$ is the thermal conductivity; $\frac{\partial T}{\partial n}$ is the temperature gradient; $\beta$ is the total heat release coefficient of structure surface, considering heat exchange of convection and radiation; $T_a$ is the air temperature in the place without illumination; $\alpha_S$ is the daily radiation heat absorption coefficient of structure surface; and $S$ is the radiation intensity.

The other method is that, when the two solids are in good contact, the temperature and heat flow at the interface are continuous, and the boundary conditions are as follows:

$$T_1 = T_2, \; \lambda_1 \frac{\partial T_1}{\partial n} = \lambda_2 \frac{\partial T_2}{\partial n} \tag{3}$$

However, if the contact between the layers is not good and the temperature is not continuous, then thermal resistance must be introduced. By establishing the assumption that the contact gap does not contain heat, the heat flow balance is maintained, and the boundary conditions are:

$$\begin{cases} \lambda_1 \frac{\partial T_1}{\partial n} = \frac{1}{R_c}(T_2 - T_1) \\ \lambda_1 \frac{\partial T_1}{\partial n} = \lambda_2 \frac{\partial T_2}{\partial n} \end{cases} \tag{4}$$

where $R_c$ is the thermal resistance caused by poor contact, $\lambda_1$ and $\lambda_2$ are the thermal conductivity of the two solids; $T_1$ and $T_2$ are the surface temperatures of the two solids; $\frac{\partial T_1}{\partial n}$ and $\frac{\partial T_2}{\partial n}$ are the temperature gradients of the two solids. The contact of the monolithic circuit board is related to the second boundary condition.

### 2.3. Radiation Intensity

The Sun continuously sends energy to the Earth in the form of electromagnetic waves. The solar constant refers to the solar radiation received per second per unit area at the top of the atmosphere perpendicular to the Sun's rays at the average distance between the Sun and the Earth. The latest observing value of $J_o$ is 1367 W/m$^2$. Considering that the distance between the Sun and the Earth changes daily, a correction coefficient for the distance between the Sun and the Earth is introduced. The intensity of solar radiation on the surface of the upper boundary of the Earth's atmosphere perpendicular to the Sun's rays is:

$$J = \xi J_o = [1.0 + 0.034 \cos(\frac{2\pi}{365}n)]J_o \tag{5}$$

where $J_o$ is solar constant; $n$ is the data number in a year; $\xi$ is the correction coefficient considering the elliptic orbit of Earth revolving around the Sun.

## 3. Results

### 3.1. Parameter Setting

Table 1 shows the material thermodynamic parameters, which are used for finite element calculations.

**Table 1.** Temperature field model material parameters.

| Parameters | Conductivity (W/(m·°C)) | Specific Heat (J/(kg·°C)) | Density (kg/m³) | Emissivity / | Film Coefficient (W/(m²·K)) |
|---|---|---|---|---|---|
| Monolithic bed board | 2.54 | 988 | 2500 | 0.94 | 13 |
| Support layer | 1.0 | 817 | 2000 | / | / |
| Soil base | 1.2 | 879 | 1870 | / | / |
| Steel | 34.9 | 520 | 7800 | / | / |

*3.2. Research Questions*

By setting the contact thermal resistance, heat flux density, and heat transfer coefficient, we simulated the heat conduction effect between different materials and the transient heat conduction effect of components such as the track bed under the action of solar radiation. We used the temperature field data of the tenth day as the calculation basis for the load–temperature coupling [28]. We introduced the temperature field data into the temperature effect calculation model and obtained the temperature stress calculation result at the corresponding time. The specific results are shown in Figure 1.

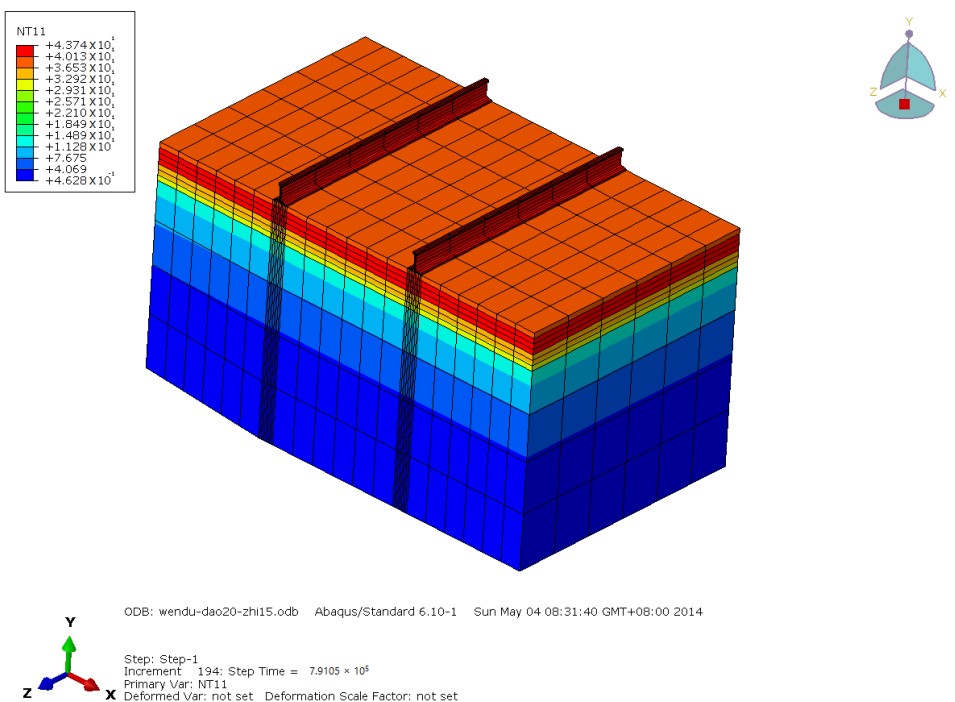

**Figure 1.** Cloud map of the temperature field calculation results of the overall track bed model.

According to the temperature deformation at different times, the most unfavorable load–temperature coupling effect can be obtained by applying the plate end load between 5:30 and 6:00 a.m. (Huang, C.W., Xie, Y., Wei, Y.Z., et al. 2016).

At the most unfavorable load–temperature coupling effect. The mechanical behavior of the overall ballast bed with different ballast slab thicknesses, support layer thicknesses, contact conditions between ballast slab and support layer, ballast slab modulus, support layer modulus and soil foundation strength were investigated. Some assumptions are as follows: It is assumed that the sides and bottom of the finite element model are both constrained by displacement. It is assumed that the joint load transfer capacity of the surface concrete slab is ignored. The structure diagram of the tramway bed is shown in Figure 2.

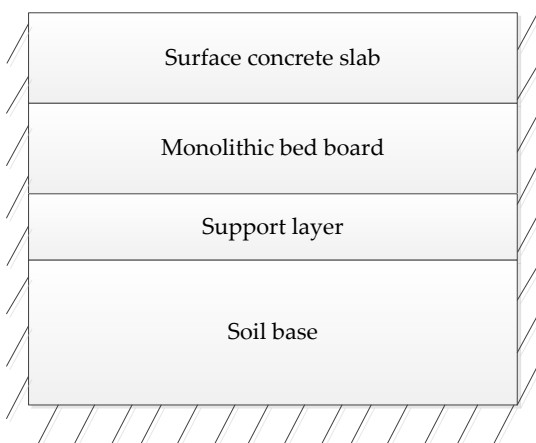

**Figure 2.** Structure drawing of tramway bed.

## 4. Discussion

### 4.1. Structure Layer Combination

#### 4.1.1. Bed Slab Thickness

We chose the monolithic track bed slabs with five thicknesses of 20, 22, 24, 26, and 28 cm to analyze the influence of different thicknesses on the strength and deflection of the track bed slab. The statistical results of the mechanical response of the bottom of the monolithic roadbed slab and the top surface of the soil foundation changing with the thickness of the roadbed slab are shown in Table 2. The horizontal tensile stress of the bottom of the track bed ($\sigma_{dy}$), the deflection of the top of the slab ($D_d$), the compressive stress on the top surface of the soil foundation ($\sigma_{sz}$), and the variation in the deflection ($D_s$) with the thickness of the track bed are shown in Figures 3 and 4.

**Table 2.** Mechanical results of different track bed thicknesses.

| Thickness of Track Bed/cm | Characteristic Points of Extreme Values of Mechanical Behavior | | | | | | | |
| | Horizontal | | | | Vertical | | | |
| | $\sigma_{dy}$/MPa | $D_d$/mm | $\sigma_{sz}$/kPa | $D_s$/mm | $\sigma_{dy}$/MPa | $D_d$/mm | $\sigma_{sz}$/kPa | $D_s$/mm |
|---|---|---|---|---|---|---|---|---|
| 20 | 2.239 | 3.301 | 66.645 | 3.387 | 2.059 | 2.719 | 100.566 | 3.965 |
| 22 | 2.069 | 3.219 | 64.205 | 3.304 | 2.133 | 2.708 | 88.990 | 3.811 |
| 24 | 1.957 | 3.126 | 62.013 | 3.210 | 2.166 | 2.705 | 82.335 | 3.655 |
| 26 | 1.841 | 3.051 | 60.764 | 3.136 | 2.158 | 2.710 | 78.769 | 3.535 |
| 28 | 1.746 | 2.977 | 59.202 | 3.062 | 2.115 | 2.718 | 77.112 | 3.427 |

In the transverse direction of the ballast plate, the tensile stress at the bottom of the ballast plate has obvious stress concentration at the wheel track. The horizontal tensile stress of the track bed slab is 2.239 MPa when the thickness of the track bed slab is 20 cm. The stress is 1.746 MPa when the thickness of the track bed slab is 28 cm. Therefore, we find that increasing the thickness of the track bed slab can significantly reduce the tensile stress level of the slab bottom. The deflection value between the two-wheel pairs at the top of the track bed slab changes greatly when the thickness of the track bed slab is constant. This finding shows that the transverse normal stress distribution in the track bed slab is relatively uneven. When the thickness of the track bed slab is constant, the deflection value slightly decreases outside the wheelset, which means that the track bed slab is pressed downward in the lateral direction under the action of the wheelset, and its deflection value decreases with the increase in the thickness of the track bed slab. Under the condition of the same thickness of the track bed, the stress basin on the top of the soil foundation is convex, and the minimum value of the lateral change is 25% relative to the maximum value. When the thickness of the track bed is 20 cm, the compressive stress on the top of the soil

foundation reaches the maximum value of 66.645 kPa, and the top of the roadbed surface deflection is 3.387 mm.

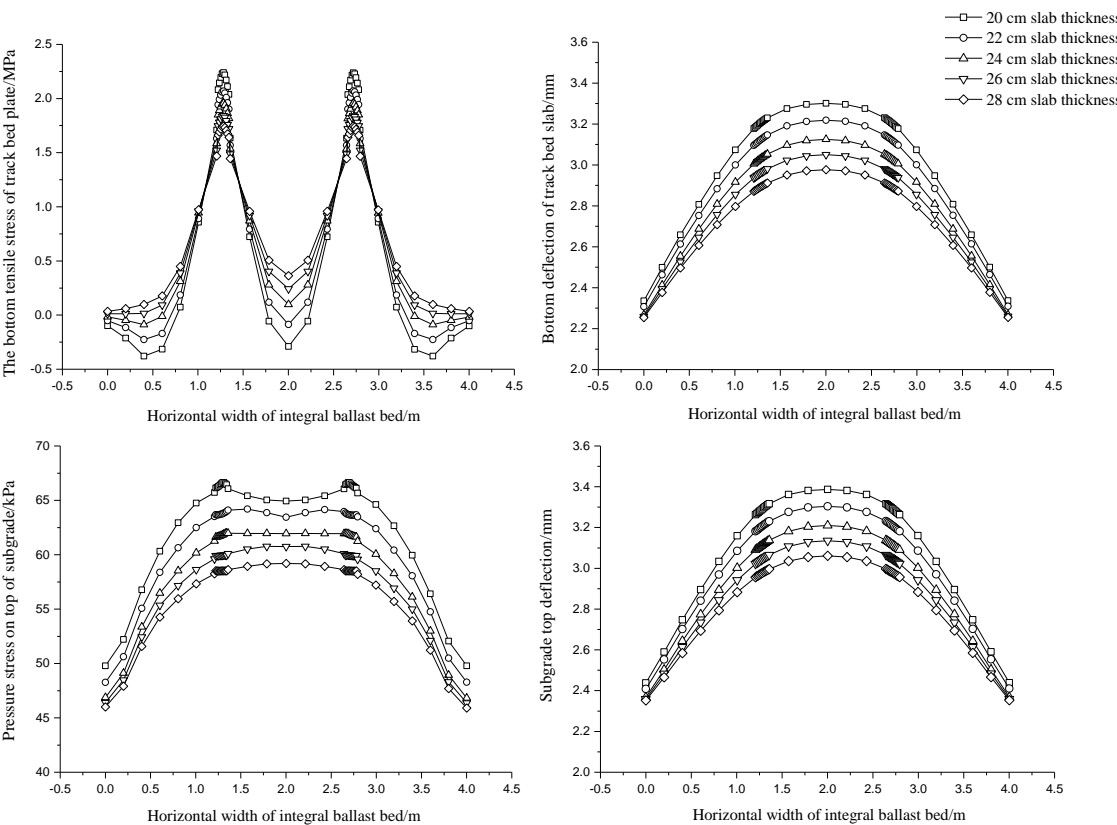

**Figure 3.** Transverse mechanics law of different track bed thicknesses.

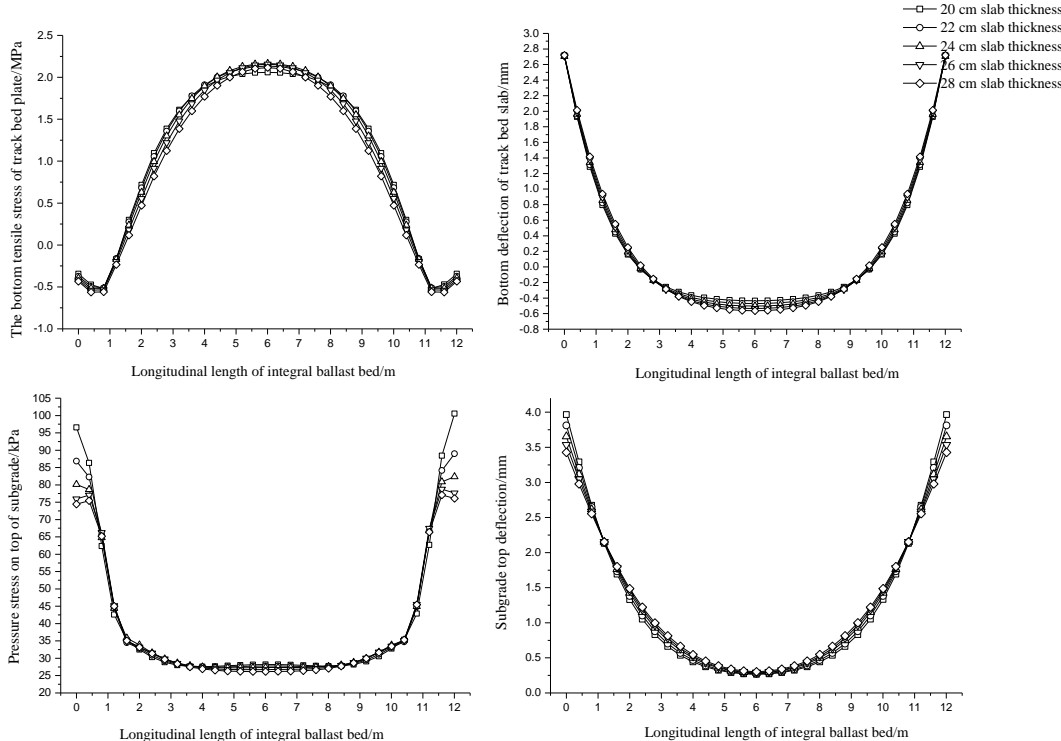

**Figure 4.** Longitudinal mechanics law of different track bed thicknesses.

In the longitudinal direction of the track bed slab, the tensile stress basin at the bottom of the slab is convex under the condition of the same track bed slab thickness, and a phenomenon of warping is observed at the end. The horizontal tensile stress at the bottom of the track bed slab changes slightly with the increase in the thickness of the track bed slab. The track bed slab is deflection and concave, but not much. In the axial direction of the track bed and the same thickness of the track bed, the stress basin on the top surface of the soil foundation is concave, the stress on the top surface of the soil foundation reaches the maximum at the end track load position, and the deflection basin is concave. When the thickness of the track bed increases from 20 cm to 28 cm, the compressive stress reduction value of the top surface of the soil foundation is approximately 23%, and the reduction value of the deflection of the soil foundation is nearly 14%.

### 4.1.2. Thickness of Supporting Layer

We selected five kinds of support layer thicknesses of 12, 14, 16, 18, and 20 cm to compare and analyze the influence of the thickness of the support layer on the strength and deflection of the track bed slab. Table 3 shows the statistical results of the mechanical response of the bottom of the slab and the top of the soil foundation with the thickness of the supporting layer. Figures 5 and 6 show the variation law of the mechanical response of the bottom of the overall track bed and the top surface of the soil foundation with the thickness of the supporting layer.

**Table 3.** Mechanical statistical results of different support layer thicknesses.

| Support Layer Thickness/cm | Characteristic Points of Extreme Values of Mechanical Behavior | | | | | | | |
| | Horizontal | | | | Vertical | | | |
| | $\sigma_{dy}$/MPa | $D_d$/mm | $\sigma_{sz}$/kPa | $D_s$/mm | $\sigma_{dy}$/MPa | $D_d$/mm | $\sigma_{sz}$/kPa | $D_s$/mm |
|---|---|---|---|---|---|---|---|---|
| 12 | 1.789 | 2.947 | 56.954 | 3.013 | 2.052 | 2.620 | 73.871 | 3.434 |
| 14 | 1.934 | 3.093 | 61.861 | 3.172 | 2.170 | 2.704 | 81.336 | 3.595 |
| 16 | 1.837 | 3.084 | 60.848 | 3.174 | 2.164 | 2.710 | 79.007 | 3.596 |
| 18 | 1.739 | 3.075 | 59.835 | 3.175 | 2.157 | 2.716 | 77.439 | 3.597 |
| 20 | 1.621 | 3.063 | 58.531 | 3.173 | 2.153 | 2.725 | 75.908 | 3.595 |

In the transverse direction of the track bed slab, the tensile stress at the bottom of the slab has obvious stress concentration at the place where the wheel tracks act. The horizontal tensile stress of the plate bottom is 1.789 MPa when the thickness of the supporting layer is 12 cm, and it is 1.621 MPa when the thickness of the supporting layer is 20 cm. When the thickness of the support layer is constant, the deflection value between the two-wheel pairs at the top of the track bed slab changes to 1.5 MPa, which indicates that the transverse normal stress distribution of the track bed slab is relatively uneven. However, the deflection value slightly decreases outside the wheelset, which implies that the track bed slab is pressed down laterally in the shape of a basin under the action of the wheelset, and its deflection value changes less as the thickness of the support layer increases. Under the condition of the same thickness of the support layer, the stress basin on the top surface of the soil foundation is convex, but the change in the thickness of the support layer does not show much regularity to the stress on the top surface of the soil foundation and the deflection of the top surface of the subgrade.

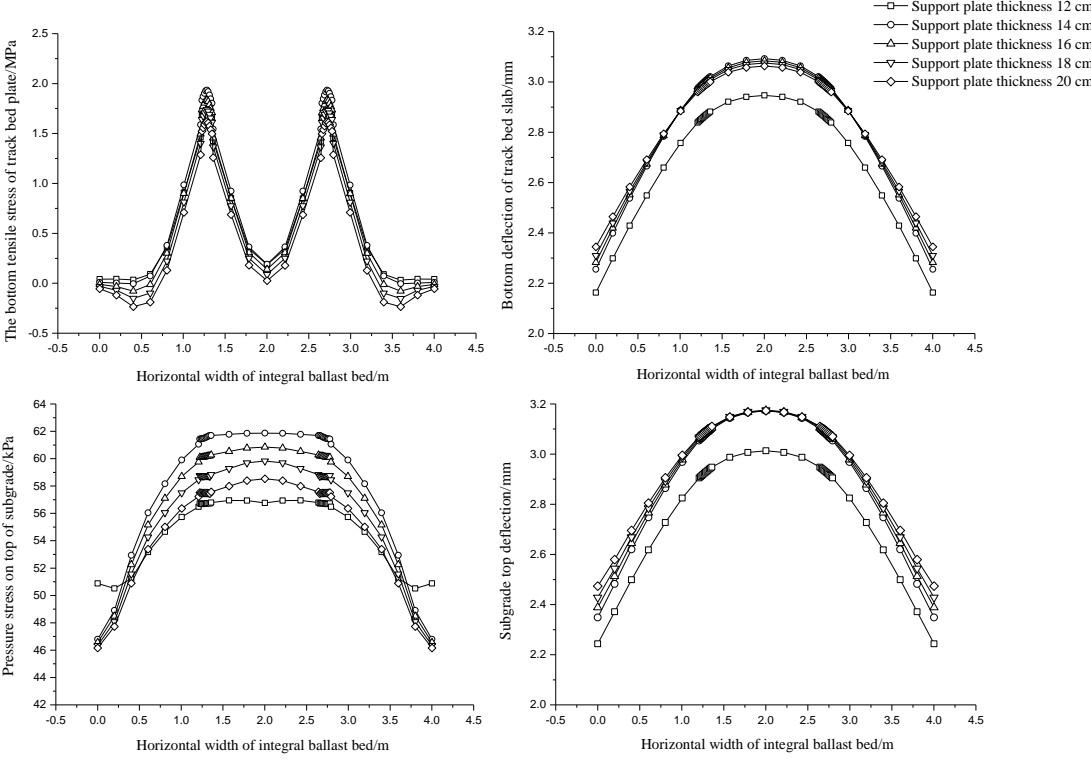

**Figure 5.** Laws of transverse mechanical behavior of different supporting layer thicknesses.

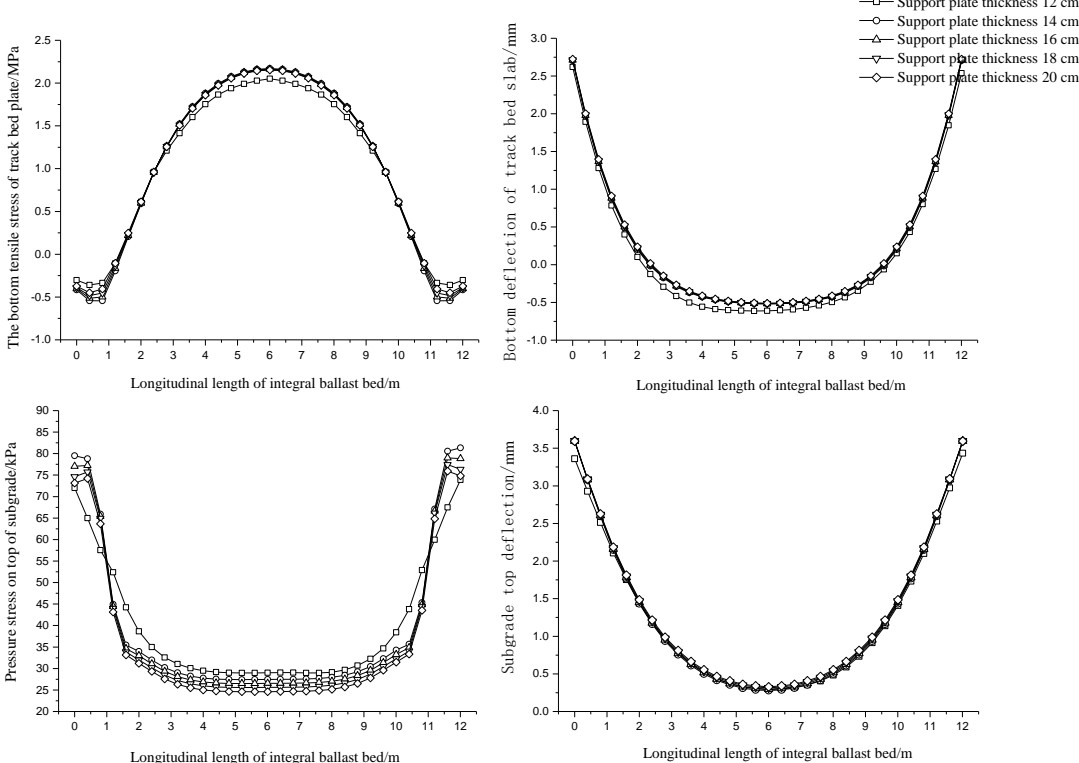

**Figure 6.** Laws of longitudinal mechanical behavior of different supporting layer thicknesses.

In the longitudinal direction of the track bed slab, the tensile stress basin of the slab bottom is convex under the condition of the same thickness of the supporting layer, and a warping phenomenon is observed at the end. The stress is 2.052 MPa when the thickness of the support layer is 12 cm, and the stress is 2.153 MPa when the thickness of the support layer is 20 cm, and the deflection of the track bed is concave. The maximum deflection is 2.620 mm when the thickness of the support layer is 12 cm. When the layer thickness is 20 cm, the maximum deflection is 2.725 m, which increases with the rise in the thickness of the support layer. In the axial direction of the track bed slab, the stress basin on the top surface of the soil foundation is concave under the condition of the same thickness of the supporting layer, the stress on the top surface of the soil foundation reaches the maximum at the end track load position, and the deflection basin is concave. When the thickness of the supporting layer is from 12 cm to 20 cm, we find that the compressive stress on the top surface of the soil foundation and the deflection of the soil foundation vary within 10%, which does not show much regularity.

### 4.1.3. Contact Conditions

According to AASHTO2020, the friction coefficient μ between the water-stabilized base layer and the surface layer is between 3.5 and 13 (mean value is 8.9) [29]. We selected five friction coefficients of 0.2, 0.6, 0.9, 1.2, and 1.5 to compare and analyze the influence of the contact conditions between layers on the strength and deflection of the track bed slab. Table 4 shows the statistical results of the mechanical response of the bottom of the monolithic slab and the top of the soil foundation changing with the contact conditions between the slab and the supporting layer. The mechanical responses of the bottom of the integral track bed and the top surface of the soil foundation vary with the contact conditions of the track bed and the support layer as shown in Figures 7 and 8. The horizontal tensile stress of the bottom of the track bed ($\sigma_{dy}$), the deflection of the top of the slab ($D_d$), the compressive stress on the top surface of the soil foundation ($\sigma_{sz}$), and the variation in the deflection ($D_s$) with the contact conditions of the slab-support layer are shown in Figures 6 and 7.

**Table 4.** Statistical results of mechanics under different contact conditions.

| Coefficient of Friction between Layers | Characteristic Points of Extreme Values of Mechanical Behavior | | | | | | | |
|---|---|---|---|---|---|---|---|---|
| | Horizontal | | | | Horizontal | | | |
| | $\sigma_{dy}$/MPa | $D_d$/mm | $\sigma_{sz}$/kPa | $D_s$/mm | $\sigma_{dy}$/MPa | $D_d$/mm | $\sigma_{sz}$/kPa | $D_s$/mm |
| 0.2 | 1.900 | 3.182 | 63.485 | 3.265 | 2.154 | 2.739 | 90.267 | 3.726 |
| 0.6 | 1.898 | 3.127 | 62.132 | 3.212 | 2.158 | 2.726 | 84.807 | 3.649 |
| 0.9 | 1.896 | 3.088 | 61.293 | 3.173 | 2.166 | 2.709 | 79.477 | 3.593 |
| 1.2 | 1.904 | 3.055 | 61.298 | 3.140 | 2.178 | 2.693 | 81.020 | 3.545 |
| 1.5 | 1.877 | 3.039 | 61.328 | 3.125 | 2.190 | 2.687 | 83.147 | 3.527 |

In the lateral position of the track bed slab, the horizontal tensile stress of the bottom of the track bed, the compressive stress of the soil foundation, and the deflection and the top of deflection slab all slightly decrease with the increase in the friction coefficient. In general, the friction coefficient only slightly affects $\sigma_{dy}$, $\sigma_{sz}$, $D_s$, and $D_d$. In the longitudinal position of subgrade slab, the horizontal tensile stress at the bottom of subgrade slab increases slightly and the compressive stress at the top of soil foundation and the deflection and the top of deflection slab decrease slightly with the increase in friction coefficient. In general, the friction coefficient only slightly affects $\sigma_{dy}$, $\sigma_{sz}$, $D_s$, and $D_d$.

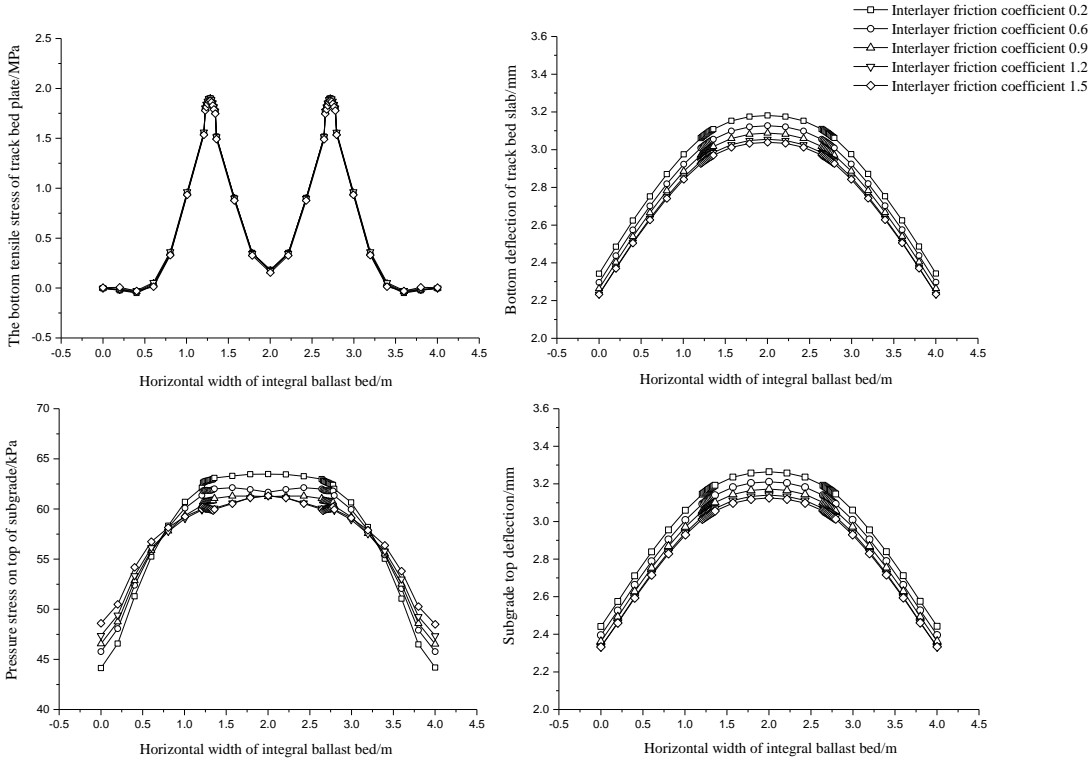

**Figure 7.** Law of lateral mechanical behavior under different contact conditions.

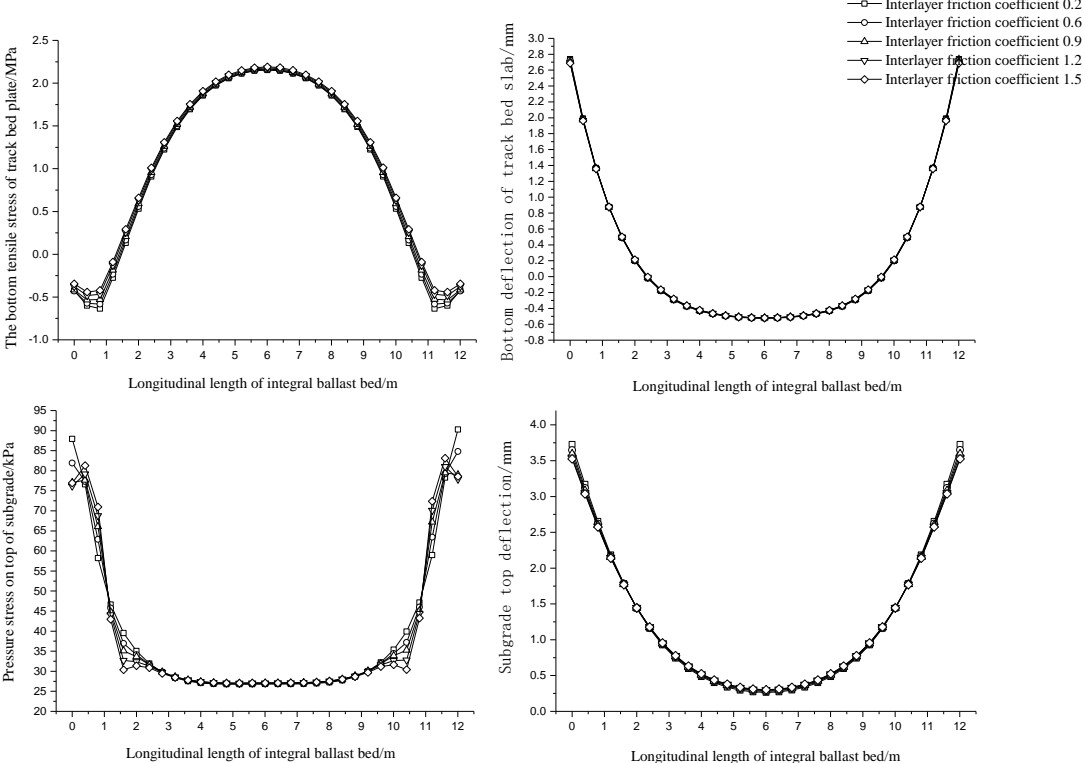

**Figure 8.** Law of longitudinal mechanical behavior under different contact conditions.

### 4.2. Material Parameters

### 4.2.1. Roadbed Plate Modulus

We selected the monolithic track bed slab with five elastic moduli of 30, 35, 40, 45, and 50 GPa to compare and analyze the influence of track bed slab modulus on the strength and deflection of the track bed slab. Table 5 shows the statistical results of the mechanical response of the bottom and top of the soil foundation of the whole bed with the modulus of the bed. The law of change is shown in Figures 9 and 10.

**Table 5.** Mechanical statistical results of different track bed slab moduli.

| Track Bed Slab Modulus/GPa | Characteristic Points of Extreme Values of Mechanical Behavior | | | | | | | |
| --- | --- | --- | --- | --- | --- | --- | --- | --- |
| | Horizontal | | | | Horizontal | | | |
| | $\sigma_{dy}$/MPa | $D_d$/mm | $\sigma_{sz}$/kPa | $D_s$/mm | $\sigma_{dy}$/MPa | $D_d$/mm | $\sigma_{sz}$/kPa | $D_s$/mm |
| 30 | 1.728 | 3.208 | 63.448 | 3.293 | 1.673 | 2.691 | 89.670 | 3.965 |
| 35 | 1.814 | 3.142 | 62.127 | 3.227 | 1.921 | 2.691 | 83.631 | 3.811 |
| 40 | 1.896 | 3.088 | 61.293 | 3.173 | 2.166 | 2.709 | 79.477 | 3.655 |
| 45 | 1.975 | 3.038 | 60.832 | 3.122 | 2.407 | 2.725 | 78.136 | 3.535 |
| 50 | 2.049 | 2.992 | 60.205 | 3.076 | 2.642 | 2.749 | 76.933 | 3.427 |

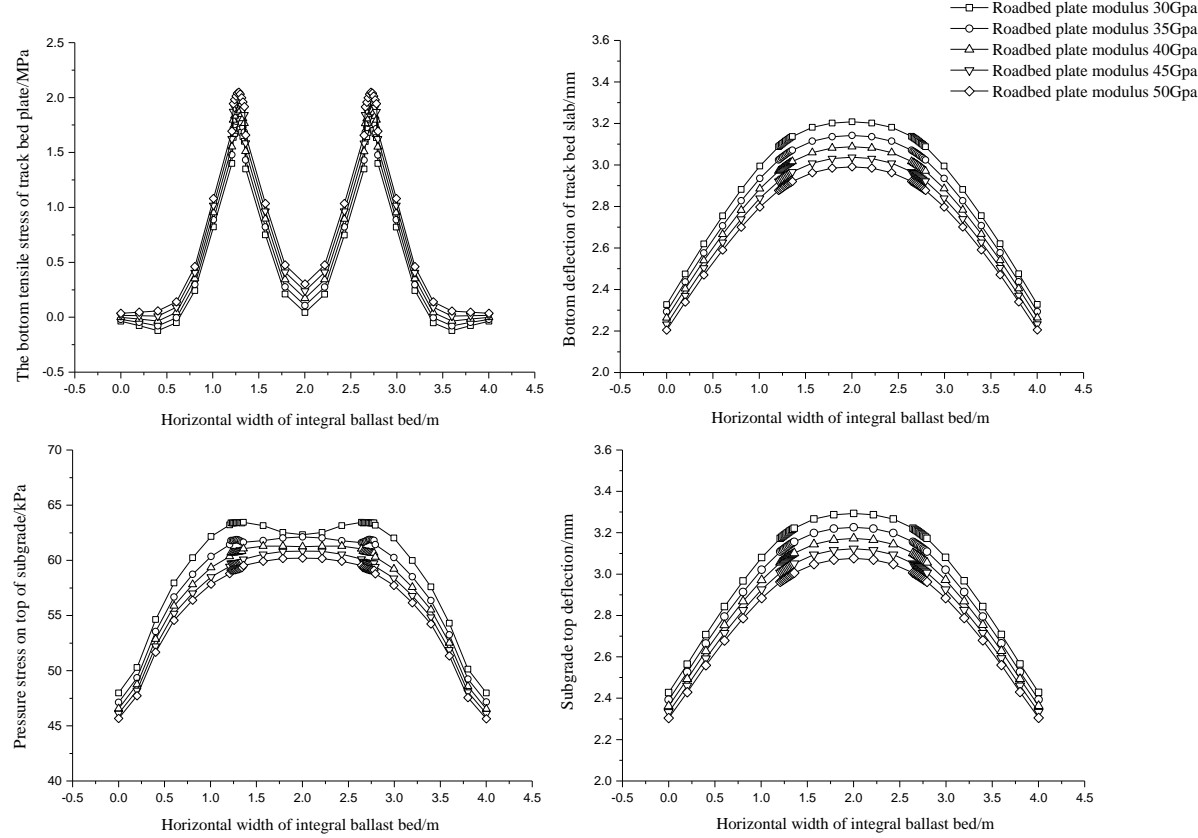

**Figure 9.** Laws of transverse mechanical behavior of different track bed slab moduli.

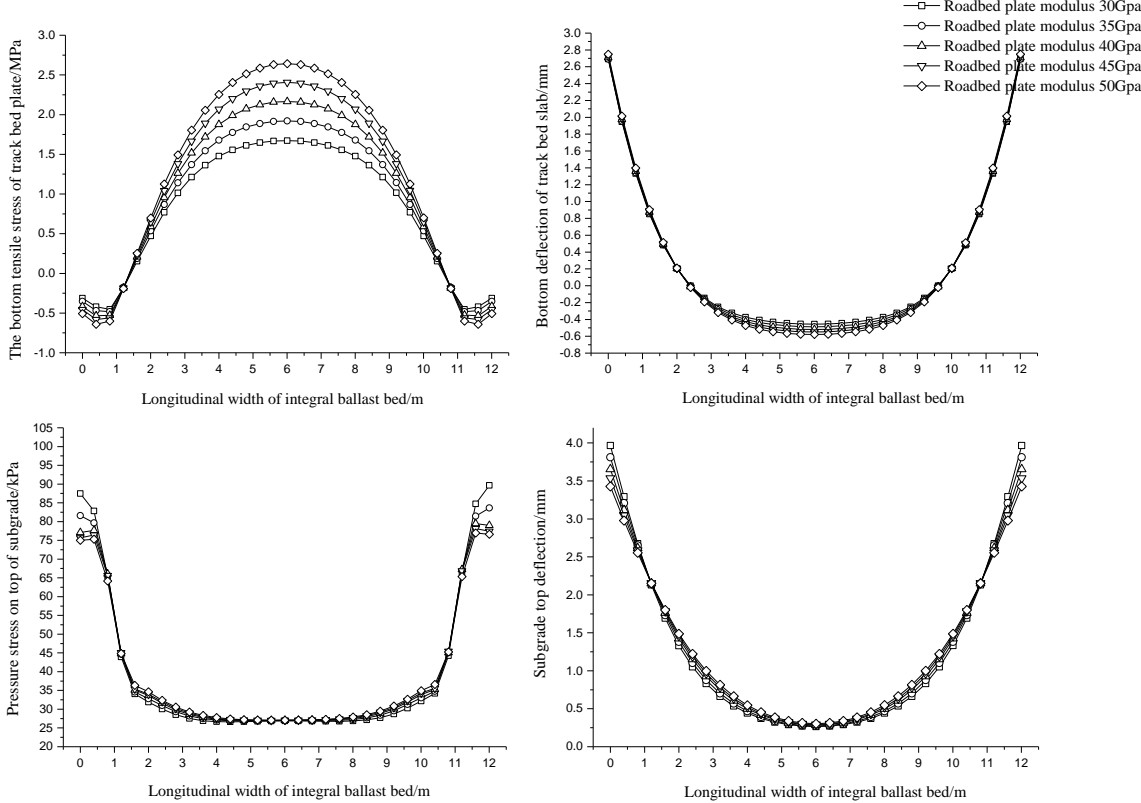

**Figure 10.** Laws of longitudinal mechanical behavior of different track bed modulus.

In the transverse direction of the track bed slab, the tensile stress at the bottom of the slab has obvious stress concentration at the place where the wheel tracks act. The stress reaches the maximum value of 2.049 MPa when the modulus of the track bed slab is 50 GPa, and it decreases as the modulus of the track bed slab decreases. The stress is 1.728 MPa when the modulus is 30 GPa. The reason is that the load component borne by the track bed slab becomes larger as the modulus of the track bed slab increases, which conforms to the stiffness distribution principle of the mechanical response of the structural layer. When the modulus of the track bed slab is the same, the deflection value between the two wheelsets at the top of the track bed slab changes to 1.7 MPa, and the deflection value decreases slightly outside the wheel set, which shows that the track bed slab is a lateral basin. The deflection value of the deflection value decreases with the increase in the thickness of the track bed slab except for the end. Under the same track bed modulus, the stress basin on the top surface of the soil foundation is convex, the minimum value of the lateral change is 28% relative to the maximum value, and the top surface of the subgrade is convex. When the modulus of the track bed is 30 GPa, the compressive stress reaches the maximum value of 63.448 kPa, and the top surface deflection of the roadbed is 3.293 mm.

In the longitudinal direction of the track bed slab and the same track bed slab modulus, the tensile stress basin at the bottom of the slab is convex, and a phenomenon of warping is observed at the end. The stress is 2.642 MPa when the modulus of the track bed slab is 50 GPa, and it is 1.673 MPa when the modulus of the track bed slab is 50 GPa. The tensile stress at the bottom of the slab increases with the rise in the modulus of the track bed slab. The track bed slab deflection is concave. The maximum deflection of the track bed slab is 2.749 mm when the track bed slab modulus is 50 GPa. When the track bed slab modulus is 30 GPa, the maximum deflection of the track bed slab is 2.691 m, which increases with the rise in the track bed slab modulus. In the axial direction of the track bed slab, the stress basin on the top surface of the soil foundation is concave under the condition of the same track bed slab modulus, the stress on the top surface of the soil

foundation reaches the maximum at the end track load position, and the deflection basin is concave. The compressive stress on the top surface of the soil foundation decreases by approximately 14% when the modulus of the track bed slab increases from 30 GPa to 50 GPa.

### 4.2.2. Modulus of Supporting Layer

We chose five kinds of support layer elastic modulus of 6, 9, 12, 15, and 18 GPa to compare and analyze the influence of the support layer modulus on the strength and deflection of the track bed slab. The statistical results of the mechanical response of the slab bottom and the top surface of the soil foundation with the change in the modulus of the supporting layer are shown in Table 6. The law of change is shown in Figures 11 and 12.

**Table 6.** Mechanical statistical results of the change in the modulus of different supporting layers.

| Support Layer Modulus/GPa | Characteristic Points of Extreme Values of Mechanical Behavior | | | | | | | |
| | Horizontal | | | | Horizontal | | | |
| | $\sigma_{dy}$/MPa | $D_d$/mm | $\sigma_{sz}$/kPa | $D_s$/mm | $\sigma_{dy}$/MPa | $D_d$/mm | $\sigma_{sz}$/kPa | $D_s$/mm |
|---|---|---|---|---|---|---|---|---|
| 6 | 2.008 | 3.119 | 60.644 | 3.119 | 2.196 | 2.709 | 85.255 | 3.628 |
| 9 | 1.952 | 3.103 | 60.333 | 3.103 | 2.180 | 2.703 | 82.208 | 3.610 |
| 12 | 1.896 | 3.088 | 61.293 | 3.173 | 2.166 | 2.709 | 79.477 | 3.593 |
| 15 | 1.821 | 3.078 | 61.367 | 3.162 | 2.154 | 2.708 | 78.886 | 3.585 |
| 18 | 1.764 | 3.065 | 61.247 | 3.150 | 2.143 | 2.712 | 78.637 | 3.572 |

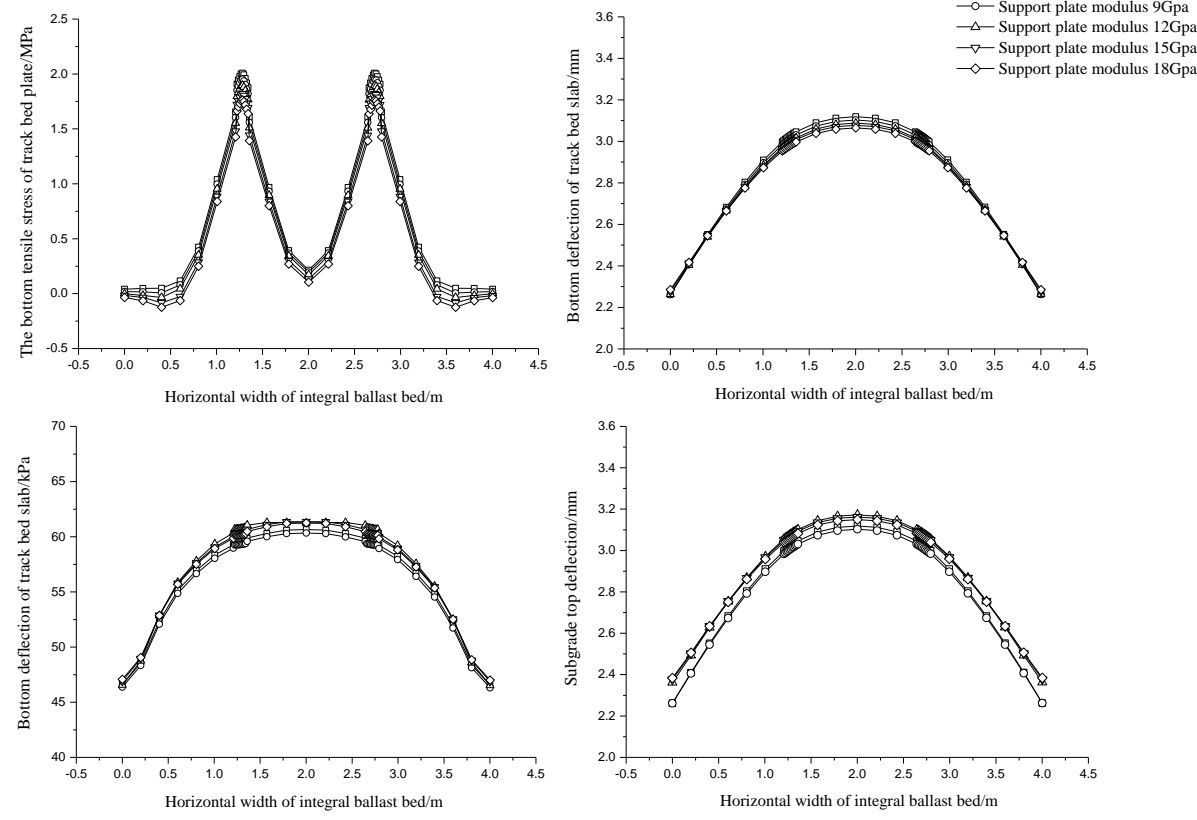

**Figure 11.** Laws of transverse mechanical behavior of different supporting layer moduli.

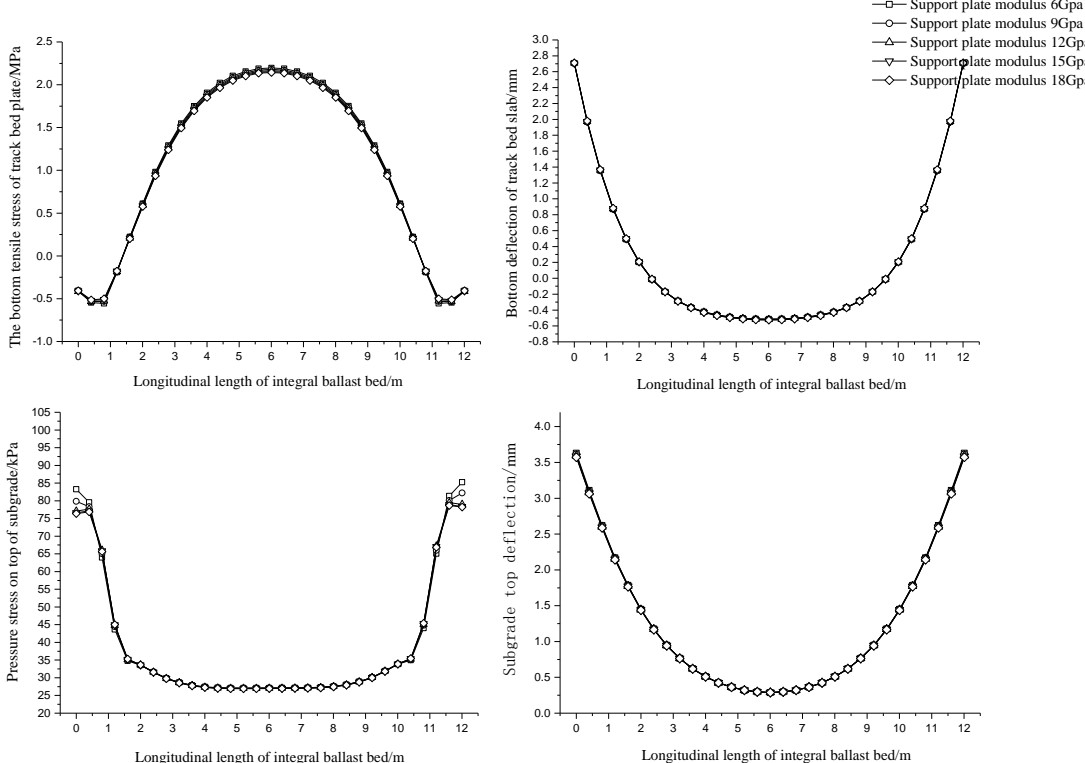

**Figure 12.** Laws of longitudinal mechanical behavior of different supporting layer moduli.

In the transverse direction of the track bed slab, the tensile stress of the slab bottom has obvious stress concentration at the place where the wheel track acts, and the horizontal tensile stress of the slab bottom decreases with the increase in the support layer modulus. When the support layer modulus is the same, the deflection value between the two wheelsets at the top of the transverse direction changes to 1.8 MPa, and the deflection value decreases slightly outside the wheelset. The increase in the modulus of the support layer decreases slightly. Under the condition of the same supporting layer modulus, the stress basin on the top surface of the soil foundation is convex, and the minimum value of the lateral change is 27% relative to the maximum value. The top surface of the subgrade is convex. When the modulus of the supporting layer is 6 GPa, the surface compressive stress reaches the maximum value of 60.644 kPa, and the top surface deflection of the subgrade is 3.119 mm.

In the longitudinal direction of the track bed slab, the tensile stress basin of the slab bottom is convex under the same condition of the modulus of the supporting layer, and a phenomenon of warping is observed at the end. The tensile stress of the plate bottom is 2.196 MPa when the supporting layer modulus is 6 GPa, and it is 2.143 MPa when the supporting layer modulus is 18 GPa. The tensile stress of the plate bottom decreases slightly with the increase in the modulus of the supporting layer. The deflection of the track bed slab is concave. The maximum deflection of the track bed slab is 2.709 mm when the support layer modulus is 6 GPa. When the support layer modulus is 18 GPa, the maximum deflection of the track bed slab is 2.712 m, and the deflection of the track bed follows the support. The layer modulus increases slightly as the layer modulus increases. In the axial direction of the track bed and the same support layer modulus, the stress basin on the top surface of the soil foundation is concave, the stress on the top surface of the soil foundation reaches the maximum at the end track load position, and the deflection basin is concave. When the support layer modulus increases from 6 GPa to 18 GPa, the compressive stress on the top surface of the soil foundation decreases by approximately 8%, and the deflection of the soil foundation decreases by nearly 2%.

#### 4.2.3. Subgrade Strength

We selected five kinds of soil foundation elastic modulus of 30, 45, 60, 75, and 90 MPa to compare and analyze the influence of soil foundation modulus on the strength and deflection of the track bed slab. The statistical results of the mechanical response of the slab bottom and the top surface of the soil foundation with the change in soil foundation modulus are shown in Table 7. The law of change is shown in Figures 13 and 14.

**Table 7.** Mechanical statistical results of different soil foundation moduli.

| Soil Foundation Modulus/MPa | Characteristic Points of Extreme Values of Mechanical Behavior | | | | | | | |
|---|---|---|---|---|---|---|---|---|
| | Horizontal | | | | Horizontal | | | |
| | $\sigma_{dy}$/MPa | $D_d$/mm | $\sigma_{sz}$/kPa | $D_s$/mm | $\sigma_{dy}$/MPa | $D_d$/mm | $\sigma_{sz}$/kPa | $D_s$/mm |
| 30 | 2.146 | 6.197 | 39.553 | 6.273 | 1.702 | 5.219 | 50.987 | 6.945 |
| 45 | 2.023 | 4.158 | 50.860 | 4.240 | 2.166 | 2.709 | 66.404 | 4.747 |
| 60 | 1.896 | 3.088 | 61.293 | 3.173 | 2.251 | 2.157 | 79.477 | 3.593 |
| 75 | 1.793 | 2.420 | 71.311 | 2.506 | 2.098 | 2.264 | 93.651 | 2.870 |
| 90 | 1.700 | 1.943 | 79.029 | 2.030 | 2.306 | 1.776 | 101.216 | 2.349 |

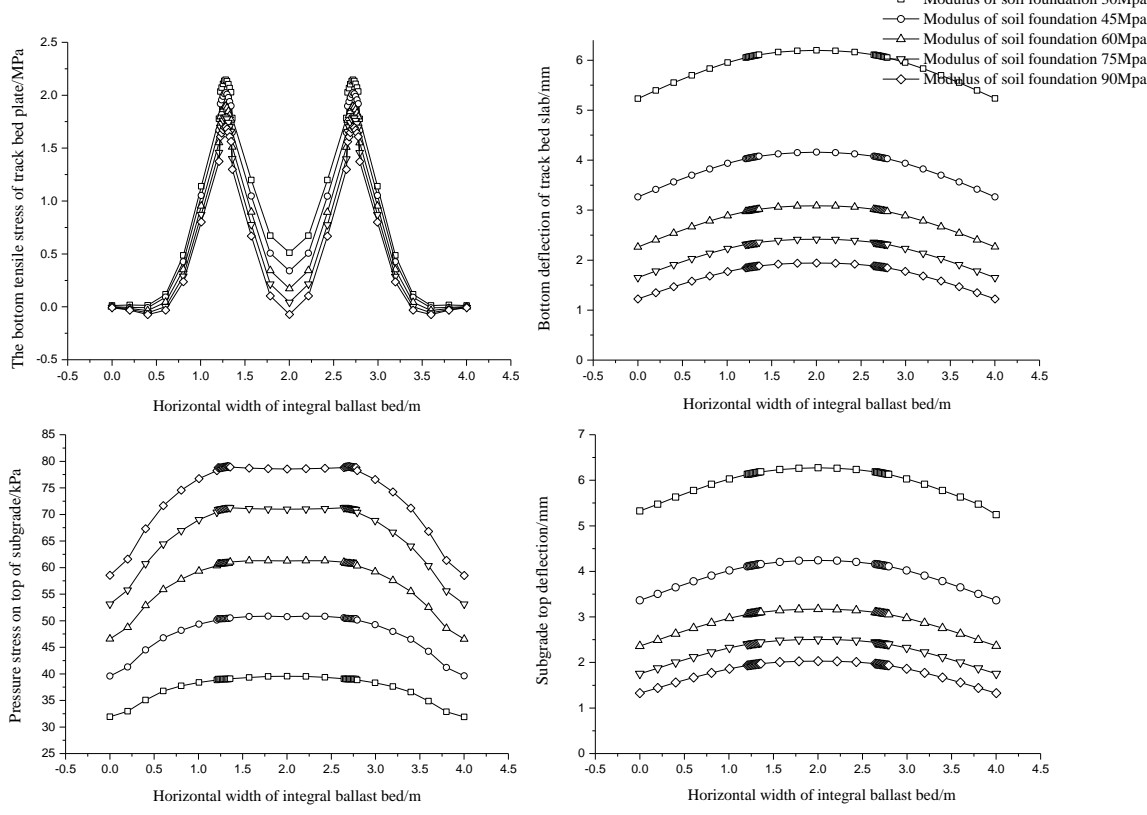

**Figure 13.** Laws of lateral mechanical behavior of different soil foundation moduli.

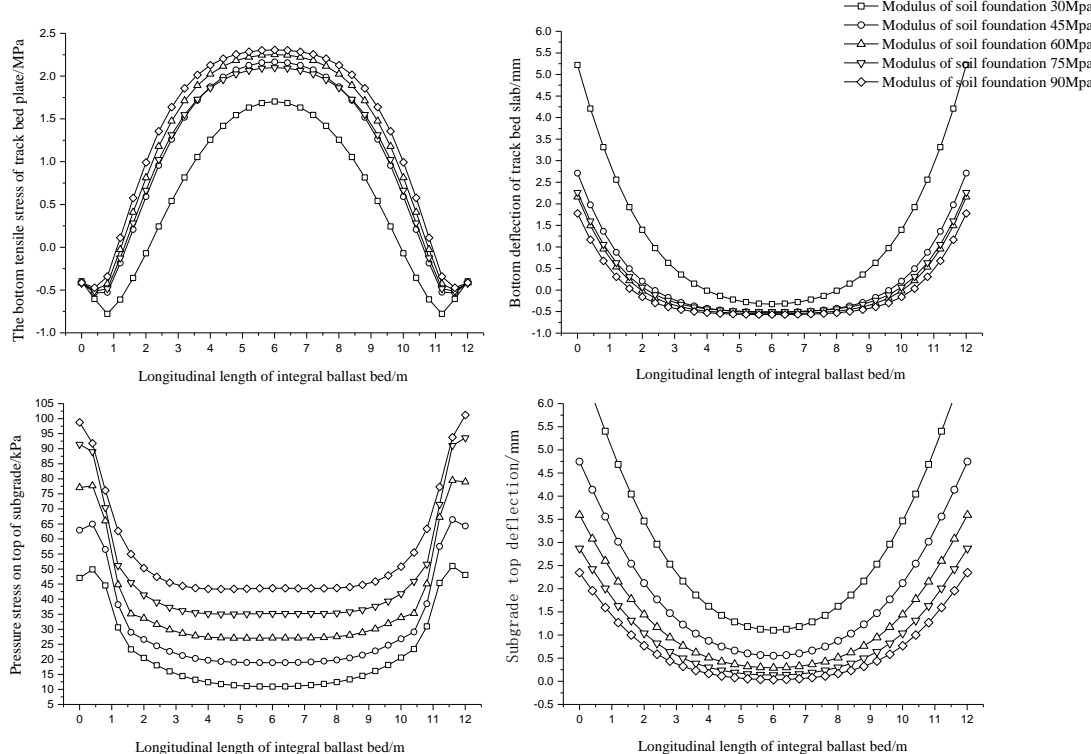

**Figure 14.** Laws of longitudinal mechanical behavior of different soil foundation moduli.

In the transverse direction of the track bed slab, the tensile stress at the bottom of the track bed slab has obvious stress concentration at the wheel track. The horizontal tensile stress of the slab bottom is 2.146 MPa when the soil foundation modulus is 30 MPa. When the modulus of the soil foundation remains unchanged, the deflection value between the two wheelsets on the top of the track bed slab changes to 1.6 MPa, and the deflection value decreases slightly outside the wheelset, which indicates that the wheelset acts. The lower track bed slab is pressed down laterally in a basin shape. When the soil foundation modulus increases, the deflection value of the track bed slab decreases sharply with the increase in the soil foundation modulus, and it converges sharply with the increase in the modulus, which indicates that the soil foundation modulus increases sharply. The deflection of the track bed slab is more sensitive when the basic modulus changes below 50 MPa. The stress basin on the top of the soil foundation is convex under the same soil foundation modulus. When the modulus of the foundation is 90 MPa, the compressive stress on the top of the foundation reaches the maximum value of 79.029 kPa, and the deflection of the top surface of the subgrade is 2.030 mm.

In the longitudinal direction of the track bed slab, the tensile stress basin at the bottom of the track bed slab is concave. The tensile stress at the bottom of the slab is 1.702 MPa when the soil foundation modulus is 30 MPa, and it is 2.306 MPa when the soil foundation modulus is 90 MPa. The tensile stress at the bottom of the track bed slab increases with the rise in the soil foundation modulus. The deflection of the track bed slab is concave. The maximum deflection of the track bed slab is 5.219 mm when the soil foundation modulus is 30 MPa, and it is 1.776 mm when the soil foundation modulus is 90 GPa. Under the same soil foundation strength condition, the stress basin on the top surface of the soil foundation is concave, and the deflection basin is concave up and down. When the modulus of the soil foundation increases from 30 MPa to 90 MPa, the compressive stress on the top surface of the soil foundation increases by approximately 99%, and the deflection of the foundation decreases by nearly 66%.

## 5. Conclusions

In this study, a finite element model is established based on ABAQUS, and it is simulated that the track bed slab reaches the maximum negative temperature difference between 5:30 and 6:00 a.m. in summer. The plate end load is applied at the maximum negative temperature difference, and the mechanical behavior of the overall ballast bed under different conditions is calculated and analyzed under the coupling action of load and temperature.

The surface of the track bed slab is pressed downward in a basin shape under the coupling effect of the load at the end of the slab and the temperature effect. The thickness change in the track bed slab can improve the lateral force and deflection of the track bed slab, but it only slightly affects the longitudinal force and deflection of the track bed slab and the longitudinal and lateral force and deflection of the soil foundation. The change in the thickness of the supporting layer has no obvious regularity to the stress on the top surface of the soil foundation and the deflection of the top surface of the subgrade. The overall law of the friction coefficient between layers is similar to that of the track bed slab, which has a weaker influence on the longitudinal and lateral forces of the track bed slab and the soil foundation. However, an appropriate increase in the friction coefficient can improve the lateral force and deflection of the track bed slab.

The increase in the modulus of the track bed slab can improve the lateral force and deflection of the track bed slab, and it has a weaker influence on the longitudinal force and deflection of the track bed slab and the longitudinal and lateral force and deflection of the soil foundation. The modulus of the supporting layer has a weak effect on the vertical and horizontal force and deflection of the track bed slab and soil foundation. However, in the engineering design, the force and deflection can be slightly improved by increasing the modulus appropriately. The modulus of soil foundation is effective in improving deformation, but it will also increase the corresponding stress. When the modulus of the soil foundation increases from 30 MPa to 90 MPa, the compressive stress on the top surface of the soil foundation increases by nearly 99%, and the deflection of the foundation decreases by around 66%.

This research has reference significance for the engineering design of the tramcar's integral track bed, which can ensure engineering quality more safely and economically. Due to limited time and space, we can further study the functional design and structural design content of the integrated track bed of trams to determine the best structural design method and load combination of the integrated track bed of trams. The research results of this study are more instructive for practical engineering, and can better reflect the concept of environmental protection and economy, which is conducive to sustainable development.

**Author Contributions:** Conceptualization: C.H. and Y.S.; methodology and software: C.H. and Y.S.; formal analysis: C.H. and M.Z.; writing—original draft preparation: C.H. and M.Z., writing—review and editing: Y.S. All authors have read and agreed to the published version of the manuscript.

**Funding:** This research was funded by Shanghai Sailing Program grant number [20YF1431900] and The APC was funded by [Science and Technology Commission of Shanghai Municipality].

**Data Availability Statement:** Not applicable.

**Acknowledgments:** This article was supported by the Shanghai Sailing Program (20YF1431900). We would also like to thank the tutors and students for their assistance in preparing the paper. We also thank the authors whose studies have been a good starting point for my research and are used as references.

**Conflicts of Interest:** The funders had no role in the design of the study; in the collection, analyses, or interpretation of data; in the writing of the manuscript, or in the decision to publish the results.

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
