# Peer review of "Load–Temperature Coupling Effect on the Base Plate End of the Whole Tram Road"

_sustainability, doi:10.3390/su14116438_

Round 1
Reviewer 1 Report
The authors analyse the temperature effects on the rail track of trams.
Although the topic is of great interest and the analytical formulation is rather exact, in the current version the paper is not eligible for publication.
Indeed, trams have been planned, designed and put into operation for more than 100 years. And, today there is certainly a consolidated methodology for defining the characteristics of the rail track.
Hence, despite the extensive analysis of the literature, it is not clear what is the current frontier of knowledge, what is the contribution of the authors and the advancement of research with respect to the state of the art.
Therefore, in order to make the contribution eligible for publication, the authors should at least:
- highlight the current frontier of research (through an appropriate analysis of the literature);
- highlight the methodological and applicative contribution proposed by the authors;
- state the main limitations of the proposed approach (limitations, simplifying assumptions, hypothesis, etc.);
- show what are the improvement margins in the case of possible future research;
- identify the case study to be analysed (for comparison) both with traditional methodologies and with the innovative approach proposed in order to highlight the improvements induced by the proposed approach.
Obviously the authors should clarify better if their proposal consists of a design methodology, a verification methodology or a solution algorithm (to solve an analytical model proposed by others).
Therefore addressing the same problem with different methodologies, or simply with different algorithms, would better understand the improvements induced by the authors’ proposal.
Finally, the authors should clarify why they publish their paper in Sustainability and not in another journal that examines, for example, the structural characteristics of materials.
Minor observations
I reckon that the authors should adopt the MDPI standard based on the use of reference number between square brackets for citing contribution in the text.
Reviewer 2 Report
The paper's main objective is the study the influence of the most unfavourable conditions of load-temperature coupling in different structures and conditions of the track bed of a tram road. A numerical analysis was performed and supported a parametric study on several variables.
The paper content is a good contribution to this subject, but some parts need more descriptions and are unclear: objectives, theory, methodology, and results.
Authors are strongly recommended to complement and clarify these parts of the manuscript. More details are provided, hoping they might be helpful.
- The objective and methodology of the paper is not well emphasized in the abstract. The abstract is more focused on the results and discussion. The reader should be more prepared firstly for the paper content.
- The abstract mentions that a finite element software was used, but the paper does not refer to the type of software.
- Are the references in the correct format? Please check if the format of the references along the text is according to MDPI template.
- The section of introduction should be clear about the innovative objectives of the paper.
- What is the reason for using the calculation methodology described in section 2.2? Explanation of some variables is missing (e.g. Ec, hc, Dc, Db, L). More details about equation 1 and its application will be helpful. A figure could help to understand some variables and the application of the methodology.
- In section 2.1, the authors mention that Tg is the maximum temperature gradient in 50 years. Why this number of years?
- The parameters of some equations should be explained. Please, check the case of equations 3 and 4.
- The authors refer (line 98) that the contact between the layers could not be good. How could be measured and expressed the degree of non-contact between layers?
- Sections 2 and 3 are too short. They present an insufficient explanation.
- Short description of the section 3 of results. More details should be presented for a better understanding of the content. It is impossible to validate the numerical analysis and results. Some descriptions are not sufficiently validated by the opinion of the authors or from the literature. The text does not make any reference to the software used by the authors in the numerical analysis. The quality of the figures should be improved (e.g. legends of Figure 1).
- The structure of the track bed is not clear along with the manuscript.
- A figure could help to visualize the parameters and the points of the structure selected for the stress analyses. See section 4.1. The explanation of these parameters is repeated in all sub-sections.
- Errors in the text when mentioning Table 2, Figure 2, and Figure 3.
- In section 4.1.3, please define the friction coefficients between layers. How were they selected? How do they represent real conditions?
- The section of conclusions does not present a short description of the paper and further studies.
- The final list of references is not organized by alphabetic order of the names.
Round 2
Reviewer 1 Report
The current version of the paper has satisfied all my previous observations
Author Response
Thank you very much for your recognition and affirmation of this article. Your previous professional guidance has been of great help to us. Looking forward to more academic exchanges and discussions with you in the future.
Reviewer 2 Report
I am not sure that the authors have provided the references in the text in the correct format. I recommend that the MDPI editorial office can confirm.
AASHTO 2020 is now a new reference of the manuscript that should be presented in the text and included in the list of references.
Author Response
Response 1: Thank you very much for your review. We have revised the citation format of the paper. We also suggest that the editorial department review the reference citation format of the paper again, and revise it in time if there are any problems.
Response 2: Thank you for your suggestion. We have supplemented the AASHTO 2020 references.